# Mining Biosynthetic Gene Clusters of *Pseudomonas vancouverensis* Utilizing Whole Genome Sequencing

**DOI:** 10.3390/microorganisms12030548

**Published:** 2024-03-09

**Authors:** Prabin Tamang, Arjun Upadhaya, Pradeep Paudel, Kumudini Meepagala, Charles L. Cantrell

**Affiliations:** 1Natural Product Utilization Research Unit, United States Department of Agriculture, University, MS 38677, USA; kumudini.meepagala@usda.gov (K.M.); charles.cantrell@usda.gov (C.L.C.); 2Department of Plant Pathology, Washington State University, Pullman, WA 99163, USA; arjun.upadhaya@wsu.edu; 3Invasive Insect Biocontrol and Behavior Laboratory, United States Department of Agriculture, 5601 Sunnyside Ave, Beltsville, MD 20705, USA; pradeep.paudel@usda.gov

**Keywords:** *Pseudomonas vancouverensis*, genome mining, natural products, antifungal, biosynthetic gene cluster

## Abstract

Natural product (NP)-based pesticides have emerged as a compelling alternative to traditional chemical fungicides, attracting substantial attention within the agrochemical industry as the world is pushing toward sustainable and environmentally friendly approaches to safeguard crops. Microbes, both bacteria and fungi, are a huge source of diverse secondary metabolites with versatile applications across pharmaceuticals, agriculture, and the food industry. Microbial genome mining has been accelerated for pesticide/drug discovery and development in recent years, driven by advancements in genome sequencing, bioinformatics, metabolomics/metabologenomics, and synthetic biology. Here, we isolated and identified *Pseudomonas vancouverensis* that had shown antifungal activities against crop fungal pathogens *Colletotrichum fragariae*, *Botrytis cinerea*, and *Phomopsis obscurans* in a dual-plate culture and bioautography assay. Further, we sequenced the whole bacterial genome and mined the genome of this bacterium to identify secondary metabolite biosynthetic gene clusters (BGCs) using antiSMASH 7.0, PRISM 4, and BAGEL 4. An in-silico analysis suggests that *P. vancouverensis* possesses a rich repertoire of BGCs with the potential to produce diverse and novel NPs, including non-ribosomal peptides (NRPs), polyketides (PKs), acyl homoserine lactone, cyclodipeptide, bacteriocins, and ribosomally synthesized and post-transcriptionally modified peptides (RiPPs). Bovienimide-A, an NRP, and putidacin L1, a lectin-like bacteriocin, were among the previously known predicted metabolites produced by this bacterium, suggesting that the NPs produced by this bacterium could have biological activities and be novel as well. Future studies on the antifungal activity of these compounds will elucidate the full biotechnological potential of *P. vancouverensis*.

## 1. Introduction

Natural products (NPs) exhibit diverse biological activities, with applications in pharmaceuticals, veterinary medicine, and crop protection in agriculture [1,2,3]. NPs are derived from various sources, such as microbes, plants, and animals, including marine organisms. Due to their structural diversity, ready degradability in the environment, and lower likelihood of inducing resistance to microbial pathogens, NPs are becoming attractive alternatives to chemical pesticides [1,3]. NPs play a role in pest management directly by killing or disrupting the pests’ life cycles and indirectly by enhancing the host resistance, e.g., Harpin Protein [4,5] and Chitosan [6]. They also serve as a foundational structure or scaffold for the discovery of novel compounds, for example, azoxystrobin derived from strobilurin [7]. 

Microbes, including bacteria and fungi, represent a remarkable source of diverse NPs or secondary metabolites. These include non-ribosomal peptides (NRPs), polyketides (PKs), ribosomally synthesized and post-transcriptionally modified peptides (RiPPs), saccharides, alkaloids, and terpenoids, which have potential in the development of novel drugs and plant protection agents [2,8,9,10,11,12]. Having evolved over billions of years, microorganisms produce diverse secondary metabolites to survive extreme conditions of the Earth as well as in their ecological niches. The genomes of these microorganisms contain the genetic instructions for synthesizing life-saving compounds, such as penicillin [13], streptomycin [14], and, in agriculture, azoxystrobin [15].

The identification, isolation, and characterization of novel NPs possessing pesticidal properties and novel mode of action (MOA) are crucial in addressing the gap left by discontinued pesticides and combating microbial resistance. However, the persistent challenge of re-discovering known compounds due to low-throughput methods has slowed down the discovery of novel compounds and requires more innovative technologies to meet the growing demand for effective pesticides [16]. Genome mining, an approach for analyzing genome sequences to identify BGCs and pathways, has emerged as a revolutionary strategy for discovering novel NPs in pharmaceutical and agriculture industries [17,18,19,20,21,22,23,24]).

With a significant decrease in DNA sequencing costs and improvements in bioinformatic tools, the whole-genome sequencing (WGS) of microbial organisms has become more frequent, enabling the rapid scanning of the genome and discovery of novel secondary metabolite BGCs. Among microorganisms, bacteria, because of their relatively simple and small genome sizes, have been the most studied for genome mining [25]. *Streptomyces* spp., known for prolific secondary metabolite production, was found to encode even more secondary metabolites through genome mining than previously predicted in the early 2000s [17,26,27]. Using WGS and bioinformatics, genome mining has unveiled many unknown metabolites from diverse bacteria: *P. koreensis*, P. fluorescens, and P. protegens [28,29,30,31,32,33,34]; *Streptomyces* spp. [35,36,37,38,39], *Burkholderia* [22,40], fungi *Aspergillus kawachii* and *Colletotrichum incanum* [23], and Lichens [24]. This underscores the versatility and potency of genome mining in exploring NPs.

Past genome mining studies on *Pseudomonas* spp. predicted several NPs. Six NRP gene clusters were reported in *Pseudomonas* sp. SH-C52, and three of them, namely thanamycin, brabantamide, and thanpeptin, were all structurally diverse lipopeptides that exhibited antifungal activity against crop fungal pathogen *Rhizoctonia solani* [28,29]. Jahanshah et al. [31] reported NRP-based orphan BGCs that encode a new lipopeptide known as gacamide A using genome mining in *P. fluorescens* Pf0-1, which has a moderate and narrow-spectrum antibiotic activity against bacterium *Arthrobacter crystallopoietes*. An endophytic bacterium, *P. fluorescens* BRZ63 collected from rape seed (*Brassica napus* L.), which has antifungal activity against crop fungal pathogens, was shown to have several genes related to antimicrobial activity such as transporters, siderophores, and other secondary metabolites [32]. A protegenin A (bacterial polyynes) was discovered using genome mining techniques from *P. protegens* Cab57, a biocontrol agent against a damping-off of cucurbits [33]. Besides being the source of natural products, *Pseudomonas* has also been demonstrated as a heterologous host for the biosynthesis of secondary metabolites as well [41]. These studies highlight the potential of discovering novel NPs from genome mining techniques. 

*Colletotrichum fragariae*, Botrytis cinerea, and Phomopsis obscurans are devastating fungal pathogens that cause anthracnose, gray mold, and leaf blight in strawberries, respectively [42,43,44]. Besides strawberries, these pathogens can cause significant yield losses in a wide range of crops, including vegetables, fruits, legumes, ornamental plants, etc. The commonly used method for managing this pathogen/disease is chemical fungicides that have negative impacts on the environment and non-targeted pathogens. Therefore, the identification and isolation of NPs that are effective against these pathogens is critical. Here, this study aimed to examine the antifungal potential of *P. vancouverenensis* against crop fungal pathogens and to mine the bacterial genome for identifying the BGCs for potential novel secondary metabolites. 

## 2. Materials and Methods

### 2.1. Bacterial Isolation and Identification

A Farkleberry (*Vaccinium arboreum*) leaf with a dark brown lesion was collected at Oxford, MS, in 2021. The lesion part was excised into ~4 cm long pieces and surface sterilized with 10% Sodium hypochlorite (NaOCl) for 2 min. Subsequently, the bleached leaf sample underwent two rinses in sterile water, each lasting 2 min, to eliminate any residual NaOCl. The washed sample was then placed onto a ½ strength potato dextrose agar (PDA) plate and incubated at 27 °C for 2 days. Bacteria that proliferated from the leaf samples were cultured onto a new ½ PDA. A pure single-colony culture was obtained through several single-colony sub-culturing steps. The obtained bacterial sample was sent for 16s rRNA sequencing to identify the bacterial species, and the sequencing was performed at GENEWIZ (Azenta Life Sciences), following their protocols.

### 2.2. Antagonistic Activity Assay against Crop Fungal Pathogens

A dual-culture plate bioassay was performed on ½ PDA media as described by Kandel et al. [45] with a slight modification. Briefly, the bacterium was grown overnight in 20 mL of tryptic soy broth (TSB) (Difco, Sparks, MD, USA) in a shaker incubator at 27 °C at 200 rpm. After incubation, the bacterial culture was centrifuged for 10 min at 5000 rpm (1968 rcf). After discarding the supernatant, the bacterial cells were washed twice with sterile TSB media and finally resuspended in 200 µL TSB media. In total, 10 µL of the bacterial suspension was spotted in 3 places on the ½ PDA plate, and a dry fungal plug (5 mm ⌀) was placed on the center of the plate to determine the antifungal activities. The observation of antifungal activities and measurement of inhibition zones were conducted seven days after incubation at 27 °C. The fungal pathogens used in this study were *C. fragariae*, *B. cinerea*, and *P. obsucurans*. Three replications were used, and the experiment was repeated twice. 

### 2.3. Bacterial Secondary Metabolite Crude Extraction

The bacterium was cultured in TSB media for three days in a shaker incubator at 27 °C at 200 rpm. Following the incubation, the bacterial culture was centrifuged for 10 min at 5000 rpm (1968 rcf). The resulting supernatant was decanted through sterile filter paper to eliminate any remaining bacterial cells. The cell-free supernatant was then subjected to ethyl acetate (EtOAc) extraction by mixing equal volumes and incubated for 5 min at room temperature. Subsequently, a separation funnel was used to separate the aqueous part from the organic solvent. The collected aqueous part was re-extracted with an equal volume of EtOAc. The obtained EtOAc extract was then dried using Rotavapor to obtain the dried crude extracts. Finally, a 20 mg/mL crude extract solution was prepared by dissolving the dried crude extract in Ethanol (95%).

### 2.4. Antifungal Bioassay

A direct bioautography, a qualitative antifungal bioassay, was conducted following the procedure outlined by Meepagala et al. [46]. Fresh conidia of *C. fragariae* and *B. cinerea* were harvested from 7–10-day-old cultures and filtered through a sterile double Mira cloth (Calbiochem-Novabiochem Crop., La Jolla, CA, USA). The spore suspension was centrifuged at 1968 rcf for 10 min, the supernatant was discarded, and the spore concentration was adjusted to 3 × 10^5^ spores/mL in PDB media (12.5 g PDB (Difco, Sparks, MD, USA), 0.5 g agar, and 0.5 mL tween 80 in 500 mL water). The bacterial crude extract (10 mg/mL) was spotted in a single volume of 10 µL (100 µg) onto a silica plate (250 microns, Silica Gel GF Uniplate, Analtech, Inc., Newark, DE, USA). After the solvent dried, the inoculum was uniformly sprayed onto a silica plate with a hand sprayer. The inoculated plate was placed in a moisture chamber box with 100% humidity and incubated at 27 ± 1 °C for four days. An MTT was sprayed in a *B. cinerea* TLC plate only to visualize the spore growth. The presence of clear zones (no fungal growth) on the TLC plate confirmed the antifungal activity of the crude extract. Technical grade fungicides captan (>98%, Chem Service, Inc., West Chester, PA, USA), fludioxonil (Analytical standard, PESTANAL^®^, Sigma-Aldrich, Inc., St. Louis, MO, USA), Carvacrol (natural, 99%, FG, Sigma-Aldrich, Inc., St. Louis, MO, USA), and thymol (Analytical 99–101%, Sigma-Aldrich, Inc., St. Louis, MO, USA) were used as positive controls. The bioassay was repeated twice.

### 2.5. Bacterial Culture, DNA Extraction, Library Construction, and Sequencing

A pure culture of *Pseudomonas vancouverensis* was grown in TSB media in a 50 mL Falcon tube for 48 h at 27 °C under continuous light in a shaker incubator at 120 rpm. Following incubation, the tube was centrifuged for 10 min at 5000 rpm (1968 rcf). The resulting supernatant was discarded, and the bacterial cells were sent to Novogene Corporation Inc., Sacramento, CA, USA, for DNA extraction, sequencing library preparation, and whole-genome shotgun sequencing (WGS) using their protocol. The bacterial DNA was extracted using the Quick-DNA Fungal/Bacterial Miniprep Kit (D6005, Zymo Research, Irvine, CA, USA), and the purity of the gDNA was assessed. A total of 200 ng high-quality gDNA was used as input material for sequencing library preparation. The WGS library was prepared using the NEBNext Ultra II kit (BioLabs, New England, Ipswich, MA, USA) according to the manufacturer’s specifications. Briefly, the gDNA was randomly sheared into short fragments (~350 bp). The DNA fragments were end-repaired, followed by A-tailing, and ligated with an Illumina adapter. The adapter-ligated fragments were PCR amplified, followed by size selection and purification. The PCR products were cleaned up using the AMPure XP bead system (Beckman Coulter, Inc., Brea, CA, USA). The quality of the sequencing library was assessed on Agilent 5400; the library showing a single major peak around 480 bp (final library size) on a bioanalyzer trace plot was considered ideal for the downstream sequencing process. The concentration of the effective library was determined through a qPCR assay. The amount of input library for sequencing was calculated based on the library concentration and the average fragment size. The qualified DNA library was sequenced on an Illumina sequencer, a Novoseq 6000 machine (Illumina, San Diego, CA, USA), with a 150 bp paired-end (PE) sequencing strategy.

### 2.6. Sequencing Read Quality Assessment and Curation

The quality of raw sequencing data was assessed using the FastQC tool (v 0.11.9) [47]. After a careful examination of the quality check (QC) reports, raw reads were processed for removing sequencing adapters, a minimum length threshold (100 bp), a minimum Phred quality score (30), and read duplication using the fastp tool (v 0.23.2) [48]. Subsequently, the high-quality reads that remained after the filtering step were subjected to normalization using the BBNorm tool of the BBMap package (v 39.01) to achieve a minimum and maximum depth of 5× and 100×, respectively [49]. This normalization step was considered essential because genome assemblers often perform poorly with both very-high- and uneven-coverage read data, resulting in a reduced efficiency or low-quality assembled genome.

### 2.7. Genome Assembly, Annotation, and Phylogenetic Analysis

The genome assembly, annotation, and phylogenetic analysis were all conducted using the comprehensive genome analysis (CGA) module available on the Bacterial and Viral Bioinformatics Resource Center (BV-BRC) web server (accessed on 20 September 2023 at https://www.bv-brc.org/app/ComprehensiveGenomeAnalysis). Briefly, for genome assembly, the parameter settings were as follows: assembly strategy = auto, pilon iterations = 2, minimum contig length = 500, and minimum contig coverage = 5. The genome was assembled in auto mode using the Unicycler assembler (v 0.4.8) [50], a SPAdes assembly optimizer. The quality of the assembled genome was assessed with QUAST (v5.2) [51] and BUSCO (v5.5.0) [52]. Gene prediction and annotation were performed with Rapid Annotation using Subsystem Technology tool kit (RASTtk 1.3.0) [53] embedded within the CGA module. For gene identification, RASTtk uses two software tools, Glimmer3 and Prodigal, and outputs the consensus prediction. Specialty genes were predicted using RASTtk through BLAT [54] search and k-mer search against the specialty gene databases, including CARD [55] and PATRIC [56] for antibiotic resistance genes, DrugBank [57]) and TTD [58] for drug targets, TCDB [59]) for transporters, and VFDB [60]) and Victors [61] for virulence factors. The comprehensive analysis performed through the CGA module ensured a detailed understanding of the assembled genome, its functional annotations, and the presence of specialized genes related to antibiotic resistance, drug targets, transporters, and virulence factors.

### 2.8. Phylogenetic Analysis

The comprehensive genome analysis (CGA) service’s default tools and techniques were employed for phylogenetic analysis and neighborhood tree construction. The Mash/MinHash technique was utilized to identify the closest relatives to the target genome. Subsequently, five global protein families shared across all genomes were selected, and their amino acid and nucleotide sequences were aligned with the MUSCLE tool. Finally, the phylogenetic relationship between the target genome and its relatives was established using the RAxML maximum likelihood method with bootstrapping, and a phylogenetic tree was constructed.

### 2.9. Secondary Metabolite Biosynthetic Gene Cluster (BGCs) Prediction

The whole-genome sequence data of *P. vancouverensis* were subjected to a prediction of secondary metabolite BGCs using the antiSMASH 7.0 bacterial version (accessed on 2 December 2023 at https://antismash.secondarymetabolites.org) with the default settings of KnownBlusterBlast, ClusterBlast, SubClusterBlast, MIBiG (Minimum Information about a Biosynthetic Gene) cluster comparison, active siteFinder, RREFinder (RiPP Recognition Element), and Cluster Pfam analysis [62,63]. The antiSMASH tool was designed to predict and annotate both existing and novel, undiscovered BGCS. 

An interactive web application, PRISM 4 (accessed on 2 November 2023 at https://prism.adapsyn.com), was used to predict the structures of the secondary metabolites produced by *P. vancouverensis* [64]. Similarly, another analysis web tool, BAGEL 4 (accessed on 5 December 2023 at http://bagel4.molgenrug.nl), was also used to determine RiPPs and bacteriocin in the bacterial genome [65]. These analyses provided insights into the potential secondary metabolites produced by *P. vancouverensis*, including the prediction of their biosynthetic gene clusters and the structural details of the metabolites. 

## 3. Results

### 3.1. Bacterial Identification and Antifungal Bioassay

The bacterium was identified as *P. vancouverensis* based on 16s RNA sequence data. In the dual-culture plate assay, the bacteria inhibited the fungal growth of *C. fragariae*, *B. cinerea*, and *P. obscurans* (Figure 1). Similarly, in the bioautography assay, the bacterial ethyl acetate extracts displayed complete antifungal activity against *C. fragariae* (Figure 2A) and *B. cinerea* (Figure 2B), which is represented by a clear white “zone of inhibition” on a TLC plate. 

### 3.2. Genome Assembly, Sequencing Statistics, and Annotation

The whole-genome shotgun sequencing of the *P. vancouverensis* genome generated a total of 17.2 million raw reads, corresponding to an estimated average genome coverage of 390× (Table 1). Following quality filtering and normalization steps, 5.9 million evenly distributed high-quality reads were retained and used for genome assembly (Table 1). The resulting draft genome of *P. vancouverensis* comprised 41 polished contigs, with a total assembly size of 6.6 Mbp, an N50 of 384 Kbp (L50 = 6), an N90 of 127 Kbp (L90 = 16), and an average GC content of 63.27% (Table 1). The largest contig in the draft genome measured 1.07 Mbp (Table 1). 

The orthologous gene search in the pseudomonadels_odb10 database on the BUSCO server revealed a 99.3% completeness of the *P. vancouverensis* genome, indicative of its high-quality sequence data (Table 1). The draft genome harbored 6052 protein-encoding genes (PEGs), 67 transfer RNA (tRNA) genes, and two ribosomal RNA (rRNA) genes (Table 1). Among 6052 PEGs, 4678 were annotated with known functions, while 1374 were classified as hypothetical proteins (Table 1). 

Based on the gene sequence homology search on specialty gene databases, the *P. vancouverensis* genome was found to harbor genes associated with antibiotic resistance, drug targets, transporters, and virulence factors (Table 2). The antimicrobial resistance genes and the virulence factor genes were found to be nearly evenly distributed across the *P. vancouverensis* genome (Figure 3A). The top five dominant sub-systems in the *P. vancouverensis* genome were found to be metabolism, protein processing, energy, membrane transport, and stress response (Figure 3B).

### 3.3. Phylogenetic Analysis

A phylogenetic analysis based on the sequences of global protein families (Appendix A) indicated that *P. vancouverensis* is closely related to gamma proteobacterium L18, followed by *Pseudomonas* sp. M47T1, *P. fragi*, *P. protegens*, and *P. fluorescens*, respectively (Figure 4). Among the *Pseudomonas* species, three species, namely *putida*, *plecoglossicida*, and *alkylphenolia* were identified as relatively distant relatives of *P. vancouverensis* (Figure 4).

### 3.4. Prediction of BGCs in P. vancouverensis

The antifungal activity exhibited by *P. vancouverensis* against various crop pathogens in this study suggests its potential to produce natural products (NPs) with antimicrobial properties. An in-silico analysis of the bacterial genome using publicly available tools, namely antiSMASH 7.0, PRISM 4, and BAGEL 4, revealed the bacterium’s capability to produce diverse types of NPs (Table 3 and Table 4, and Figure 5).

#### 3.4.1. Prediction of NP BGC with antiSMASH

Out of a total of 41 contigs of the bacterial genome, antiSMASH 7.0 predicted a total of 17 BGCs on 15 contigs, where contigs 1, 4, 5, 6, 7, 8, 10, 12, 14, 22, 26, 31, and 34 each harbored a single BGC, whereas contigs 2 and 11 each contained two BGCs (Table 3). The remaining 26 contigs did not contain regions with potential secondary metabolites. Among the 17 BGCs, 11 were annotated as NRPs or NRP-like genes while the remaining 6 were annotated as NAGGN, redox-cofactor, RiPP-like, aryl polyene, hydrogen cyanide, or CDPS-type genes (Table 3). Ten BGCs identified in the *P. vancouverensis* genome shared homology with known BGCs, including bovienimide A (NRP: Lipopetide, 100% similarity), pyoverdine (NRPS-like, 7% similarity), lankacidin C (redox-cofactor, NRP + Polyketide, 13% similarity), arylpolyene (APE Vf, 40% similarity), frederiksenibactin (NRPS, hserlactone, 15% similarity), pyoverdine SMX-1 (NRP-Metallophore, 35%), Pf-5 pyoverdine (NRP-metallophore, 9% similarity), and nostopeptolide A2 (Polyketide + NRP: Cyclic depsipeptide, 37% similarity) (Table 3). Bovienimide A is an NRPs-type secondary metabolite that has been reported to be produced by the bacteria *Xenorhabdus bovienii* [38]). Other clusters like N-acetylglutaminylglutamine amide (NAGGN), RiPP-like, hydrogen-cyanide (HCN), and cyclodipeptides (CDPs) did not match with previously reported BGCs. This result indicates that *P. vancouverensis* has a significant potential for producing novel secondary metabolites. 

#### 3.4.2. Prediction of NP BGC with PRISM4

Additionally, the PRISM 4 algorithm was used to predict BGCs and their structures from the *P. vancouverensis* genome, and the result can be found at https://prism.adapsyn.com/results/b0179e980bba9b6351dcc43caa5d9b89 (accessed on 2 November 2023). A total of 12 BGCs were identified within the ten contigs of the *P. vancouverensis* genome (Table 4). Among the 12 BGCs, 8 were annotated as NRPs, 2 as PKs, 2 as acyl-homoserine lactone, and 1 as cyclodipeptide (XYP family) (Table 4). Within these clusters, a total of 14 NPs had predicted structures (Figure 4) with their IUPAC nomenclature (International Union of Pure and Applied Chemistry) (Appendix A). Based on the chemical structures, 12 NPs were distinct, while 2 NPs in clusters 4 and 12 were identical (Figure 4).

#### 3.4.3. Prediction of NP BGC with BAGEL4

The BAGEL 4 analysis tool identified one cluster associated with RiPPs or bacteriocin in the *P. vancouverensis* genome. This cluster contained 30 putative genes, of which 5 had functional annotations (Table 5). Notably, a putidacin_L1 family-like bacteriocin, a class III bacteriocin with molecular weight >10 kDa, was among the predicted secondary metabolites (Table 5).

## 4. Discussion

In agriculture, NP-based pesticides are recognized as a primary alternative to chemical fungicides, drawing significant interest from the agrochemical industry [1,2,66,67]. Microbes, encompassing both bacteria and fungi, are untapped sources of NPs with valuable applications in the pharmaceutical, agriculture, and food industries [68,69,70,71]. Many microbial metabolites remain undiscovered using traditional low-throughput methods due to the association of NPs with cryptic genes, which often require external stimulation for microbes to produce [16]). Genome mining facilitates the identification of those cryptic, uncharacterized BGCs responsible for synthesizing NPs, which are likely novel [18,72].

Molecular identification using 16s rRNA sequencing is a commonly used molecular tool for identifying bacteria at the genus level and sometimes at the species level with a 65–85% accuracy [73], and we utilized it to determine the bacterium used in this study. The bacterial species isolated in this study was confirmed as *P. vancouverensis* based on the 16s rRNA sequence data. The phylogenetic analysis result based on WGS data further confirmed the identification of this bacterium, and it is closely related with gamma proteobacterium and *Pseudomonas* sp. M47T1. The first strain of *P. vancouverensis*, a Gram-negative bacterium, was isolated from forest soil in Vancouver, Canada [74]. *Pseudomonas* are reported to be predominantly in soil and are aquatic environment inhabitants [75]. However, we isolated *P. vancouverensis* from the plant Farkleberry (*V. arboreum*). Mishra et al. [76] also isolated this species from a garlic (*Allium sativum*) rhizosphere, confirming our results that this species can also be endophytic. This bacterial species has been reported to control fire blight (*Erwinia amylovora*) in apples and pears [77], and several other species of *Pseudomonas* have also been shown to have antifungal activity against several crop fungal pathogens [78]. In our study, this bacterium exhibited antifungal activity against crop fungal pathogens *C. fragariae*, *B. cinerea*, and P. obscurans in an in-vitro bioautography and dual-culture plate assay. Both the bioautography and dual culture were qualitative assays.

The organic compound extract was subjected to a bioassay-guided fractionation to identify the bioactive antifungal compounds from this bacterium. However, due to the instability or volatility of the compounds, the active constituents were lost consistently during the isolation process. We will follow up on the bioassay-guided fractionation of the bioactive compounds from this bacterium in the future. Since the in vitro assays have clearly demonstrated the antifungal activity of the bacteria, it could be a good candidate for biological control agents. However, further assessment is needed to better understand its potential impact on human health and the environment.

In this study, the genome mining of *P. vancouverensis* identified several biosynthetic gene clusters (BGCs), predominantly predicted to produce NRPs or NRP-like metabolites, as Saati-Santamaria et al. [79] reported. All predicted BGCs shared less than a 50% identity with previously reported gene clusters except for one BGC (NRP-like) that was 100% identical to bovienimide-A, a synthetase produced by *Xenorhabdus bovienii* [38] (Table 3). The findings from our study suggest that the *P. vancouverensis* genome harbors a repertoire of novel BGCs that potentially encode for undiscovered bioactive natural compounds, and thus warrants detailed investigation. *X. bovienii* is an entomopathogenic bacterium that has a mutualistic relationship with the nematodes *Sterinernema* sp.; however, it has not been reported if bovienimide-A produced by *X. bovienii* has an insecticidal activity [38]. The bovienimide-A predicted in *P. vancouverensis* could represent a novel secondary metabolite regarding the fungicidal property, but it needs to be examined. 

Pyoverdines, siderophores that bind iron, are produced by specific *Pseudomonads* and play a vital role in the virulence and pathogenicity of these bacteria. These compounds act as bacterial-secreted toxins, disrupting mitochondria and iron homeostasis in the free-living transparent nematode *Caenorhabditis elegans* [80,81]. Similarly, viobactin, a predicted non-ribosomal peptide (NRP) compound in *P. vancouverensis*, is also linked to iron uptake and virulence in the bacterium *Chromobacterium violaceum* [82]. Frederiksenibactin, another predicted compound, is also associated with the Fe (III) complex. Considering that iron is an essential element for the growth of nearly all living organisms, we hypothesized that the pyoverdine, viobactin, and frederiksenibactin predicted to be produced by *P. vancouverensis* might have exerted the antifungal activity against the crop pathogens used in the dual-culture plate assay.

Some *Streptomyces* species have been reported to produce lankacidin C, a polyketide/NRP-class natural product. Known for its intricate and varied structures, this compound has showcased diverse biological activities, including antibacterial and antitumor properties [83,84]. Notably, lankacidin C was shown to exhibit antimicrobial activity through the inhibition of bacterial ribosomes [84]. The ribosome is a vital cellular machinery involved in protein synthesis. Thus, the lankacidin C predicted in the *P. vancouverensis* could have an antifungal property against crop pathogens with a similar MOA as reported in the literature.

Hydrogen cyanide (HCN), a BGC predicted in *P. vancouverensis*, did not match with the known BGC database; however, the HCN compound produced by certain *Pseudomonas* spp. has been found to exhibit an antifungal property [85,86]. Michelsen and Stougaard [85] reported that the *P. fluorescens* strain In5 produced HCN and inhibits the growth of plant fungal pathogens Rhizoctonia solani and Pythium aphanidermatum. In a separate study [86], reported the antifungal activity of *Pseudomonas* spp. against potato late blight (Phytophthora infestans), which was associated with HCN. So, we hypothesized that the HCN predicted in *P. vancouverensis* might have been attributed to the antifungal properties against crop pathogens.

A few of the BGCs predicted in this study, such as nostopetolide A2 and aryl polyenes, have not been reported to possess antimicrobial or antifungal properties to date. Nostopeptolide has been isolated from certain strains of cyanobacteria, belonging to the genus *Nostoc* [87], and aryl polyenes are a prevalent class of bacterial NPs. However, some polyene compounds are known for their antifungal properties [88,89,90].

RiPPs and bacteriocin are important classes of secondary metabolites often associated with antimicrobial properties in prokaryotes [91,92,93]. Bacteriocins, traditionally utilized as food preservatives, are gaining increased consideration for their potential as antimicrobial agents [91,93]. Hence, the genome of *P. vancouverensis* was subjected to a computational analysis to identify regions containing these metabolites. We identified a BGC linked to bacteriocin biosynthesis, where a putidacin_L1 family-like bacteriocin (a class III bacteriocin) was identified as the core peptide. Interestingly, a comparative analysis of the publicly available genomes of other *P. vancouverensis* strains revealed the absences of RiPPs and bacteriocin in those strains, suggesting a strain-specific secondary metabolite. Rooney et al. [94] have reported the antibacterial properties of putidacin against *P. syringae* pv. *syringae* by expressing it in *Nicotiana benthamiana*. So, putidacin might have contributed antifungal properties against the fungus used in this study, and further study will determine the antifungal activity of putidacin against the crop pathogens used in this study. 

This study provided comprehensive insights into the genomic features and functional elements present in *P. vancouverensis*. Overall, the in-silico analysis suggests that *P. vancouverensis* possesses a rich repertoire of BGCs with the potential to produce diverse and novel NPs, including NRPS, PKS, acyl homoserine lactone, cyclodipeptide, bacteriocins, and RiPPs. The results show the capability of this bacterium to produce a diverse and large number of NPs that may have potential antibiotic or antimicrobial activity against agricultural crop fungal pathogens, making this bacterium an excellent microbial candidate for mining novel NPs. Decoding microbial genomes and harnessing their NPs can address critical challenges in healthcare and agriculture, contributing to food security, and a healthier and more sustainable future.

## Figures and Tables

**Figure 1 microorganisms-12-00548-f001:**
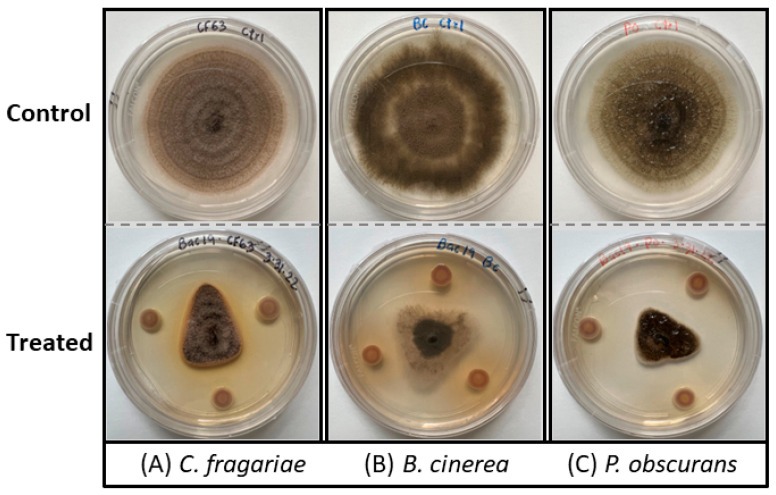
Dual-culture plate assay showing the antifungal activity of *P. vancouverensis* against plant fungal pathogens: (**A**) *C. fragariae*, (**B**) *B. cinerea*, and (**C**) *P. obscurans*. The fungal growth was compromised in the *P. vancouverensis*-treated plate as compared to the control plates. The top row is the control, fungus only, and the bottom row was *P. vancouverensis*-treated (fungus + bacteria).

**Figure 2 microorganisms-12-00548-f002:**
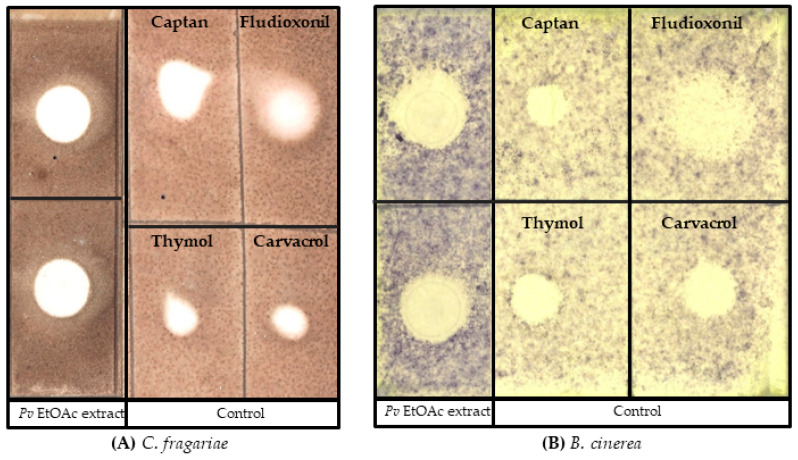
Bioautography assays depicting the antifungal activity of *P. vancouverensis* EtOAc extract. A total of 10 µL (100 µg) EtOAc extract was spotted onto a silica gel plate. The clear “zone of inhibition” represents the antifungal activity of the crude extract against the fungal pathogens (**A**) *C. fragariae* and (**B**) *B. cinerea*.

**Figure 3 microorganisms-12-00548-f003:**
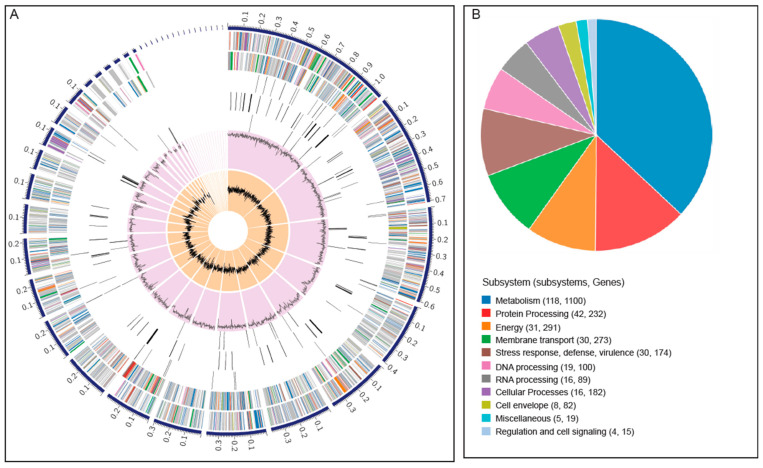
(**A**) Circos plot depicting the distribution of genome annotations in the *Pseudomonas vancouverensis* genome. From the outer to inner rings: contigs, CDS on the forward strand, CDS on the reverse strand, RNA genes, CDS with similarity to known antimicrobial resistance genes, CDS homologous to known virulence factors, GC content, and GC skew. The numbers on each contig (outermost ring) indicate physical positions in Mbp. The color bands of CDS on forward and reverse strands indicate different sub-systems that those genes are predicted to be involved in. The code for each color is in the legend of the adjacent pie chart. (**B**) Pie chart displaying the distribution of sub-systems in the *P. vancouverensis* genome.

**Figure 4 microorganisms-12-00548-f004:**
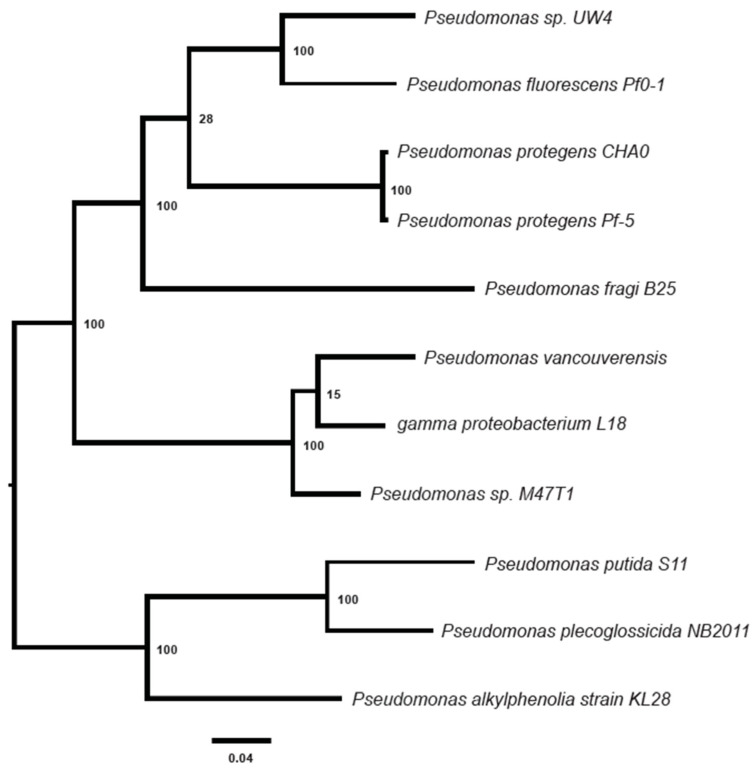
A maximum likelihood phylogenetic tree illustrating the evolutionary relationship between *Pseudomonas vancouverensis* and other *Pseudomonas* species. Support values at each branch node represent confidence levels through 100 rounds of bootstrapping. The horizontal scale bar indicates the amount of genetic change, measured as nucleotide substitutions per site along the branches.

**Figure 5 microorganisms-12-00548-f005:**
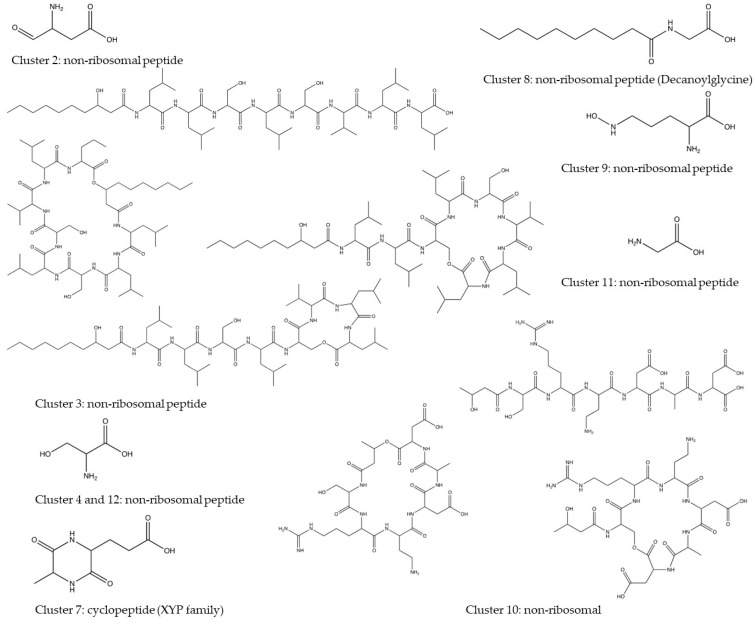
Predicted structures of secondary metabolites produced by *P. vancouverensis* using the PRISM4 prediction tool.

**Table 1 microorganisms-12-00548-t001:** Sequencing statistics, assembly statistics, genome features, and protein features predicted in the genome of *Pseudomonas vancouverensis*.

Features ^a^	*P. vancouverensis*
**Sequencing statistics**	
Number of raw sequencing reads	17,268,958
Average length of raw reads (bp)	150
Average coverage with raw reads (x)	390
Quality sequencing reads	14,402,604
Reads retained after normalization	5,989,222
**Assembly statistics**	
Number of contigs	41
Total length (bp)	6,646,214
Largest contig (bp)	1,079,378
N50 (bp)	384,424
N90 (bp)	127,873
L50	6
L90	16
GC (%)	63.27
Busco completeness (%)	99.3
**Genome features**	
CDS	6052
tRNA	67
rRNA	2
Partial CDS	0
Misc. RNA	0
Repeat Regions	0
**Protein features**	
Protein-encoding genes (PEGs)	6052
PEGs with functional assignment	4678
Hypothetical proteins	1374
Proteins with GO assignments	1148
Proteins with pathway assignments	1018

^a^ CDS, coding sequence; tRNA, transfer RNA; rRNA, ribosomal RNA; Misc., miscellaneous, GO, gene ontology.

**Table 2 microorganisms-12-00548-t002:** Number of specialty genes predicted in the *Pseudomonas vancouverensis* genome.

Specialty Genes	Database	Number of Genes
Antibiotic Resistance	CARD	3
	PATRIC	85
Drug Target	DrugBank	24
	TTD	6
Transporter	TCDB	59
Virulence Factor	VFDB	27
	Victors	18

**Table 3 microorganisms-12-00548-t003:** Putative gene clusters of secondary metabolites of *P. vancouverensis* using antiSMASH 7.0.

Region	Gene Type ^a^	Span (nt)		Most Similar BGCs	Type	Similarity ^b^
		From	To			
1.1	NAGGN	665,212	679,985	-	-	-
2.1	NRPS-like	242,243	285,626	Pyoverdine	NRP	7%
2.2	redox-cofactor	612,182	634,341	Lankacidin C	NRP + Polyketide	13%
4.1	RiPP-like	63,468	75672	-		
5.1	NRPS	179,999	224,264	Viobactin	NRP	15%
6.1	NRPS	190,093	257,271	MA026	NRP	10%
7.1	Aryl polyene	36,509	80,104	APE Vf	Other	40%
8.1	Hydrogen-cyanide	225,474	238,306	-	-	-
10.1	NRPS, hserlactone	119,551	178,652	Frederiksenibactin	NRP	15%
11.1	hserlactone	51,665	72,261	-	-	-
11.2	CDPS	77,532	98,269	-	-	-
12.1	NRPS	185,164	229,126	-	-	-
14.1	NRP-metallophore, NRPS	143,663	182,106	Pyoverdine SMX-1	NRP	35%
22.1	NRP-metallophore, NRPS	1	38,911	Pf-5 pyoverdine	NRP	9%
26.1	NRPS	1	4959	Nostopeptolide A2	Polyketide + NRP: Cyclic depsipeptide	37%
31.1	NRPS	1	2237	-	-	-
34.1	NRPS-like	1	1903	Bovienimide A	NRP:Lipopeptide	100%

^a^ NRPS—Non-ribosomal peptide synthetase. ^b^ The “similarity” is the percentage of homologous genes in the query and hit clusters.

**Table 4 microorganisms-12-00548-t004:** Putative gene clusters of secondary metabolites of *P. vancouverensis* using PRSIM 4.

Gene Cluster Type	Contigs	Clusters	Predicted Structure
PK	1	1	No
NRP	2	2	Yes
NRP	3	3	Yes
NRP	4	4	Yes
Acyl Homoserine Lactone		5	No
Acyl Homoserine Lactone	5	6	No
Cyclopeptide (XYP family)		7	Yes
NRP	6	8	Yes
NRP	7	9	Yes
NRP	8	10	Yes
NRP	9	11	Yes
NRP	10	12	Yes

**Table 5 microorganisms-12-00548-t005:** RiPP and bacteriocin predicted by BAGEL 4.0. Only a blast hit with UniRef90 is presented.

Areas of Interest	Annotation
Orf00003	Large-conductance mechanosensitive channel OS = *Pelobacter propionicus*
Orf00011	HTH-type transcriptional regulator for conjugative element SXT OS = *Vibrio cholerae*
Orf00019	Uncharacterized protein HI_1412 OS = *Haemophilus influenzae*
Orf00026	P21 prophage-derived protein NinB OS = *Escherichia coli* O6:H1
Orf00036	Putidacin_L1 family lectin-like bacteriocin *

* This protein was identified only in this *P. vancouverensis* and absent in other strains of *P. vancouverensis* present in the NCBI database.

## Data Availability

The data presented in this study are available in the article. The genomic data were deposited into NCBI database under project ID #PRJNA1071670 and accession #SAMN39708908.

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
