# Peer review of "Mining Biosynthetic Gene Clusters of Pseudomonas vancouverensis Utilizing Whole Genome Sequencing"

_microorganisms, 2024, doi:10.3390/microorganisms12030548_

Round 1

Reviewer 1 Report

Comments and Suggestions for Authors

The authors of this manuscript aimed to examine the antifungal potential of P. vancouverensis. Additionally, the authors mined its genome to identify BGCs with may possess novel secondary metabolites of interest.

While I find the idea behind this study of relevance I am not particularly satisfied with the story provided. The use of bioinformatics tools coupled with experimental validation are essential in creating links between what is predicted and what in reality happens. In this study the authors did a very good job experimentally to assess antimicrobial properties against certain pathogens but did not validate any of the NPs predicted by bioinformatics tools. Too many hypotheses with very little to back them up. I think that confirming at least 1 or 2 of the predictions would strengthen the manuscript substantially.

I would also restructure slightly the results section. Maybe create subsections for the use of the different tools so the reader doesn't get lost.

Minor corrections in the text and formatting issues:

Line 46: (e.g., Harpin Protein) (Chuang et al., 2014; Liu et al., 2020); and 46 Chitosan (Riseh et al., 2022).

Shouldn’t the parentheses highlighted in red be placed at the of the sentence? From what I understood, two examples are given form host resistance: Harpin protein and Chitosan.

Page 9: The line numbers are overlapping with Table 1.

Figure 3 legend: “antimicro-bial” – remove the dash. Also, what are the numbers on the circus plot referring to? (e.g., 0.1). Please add the explanation to the legend.

Figure 4: Please add an explanation to the different numbers.

Legend Figure 5: Font size too big. Please make fonts consistent.

Line 417-419: Phrase construction is a bit confusing (e.g. while … while). Perhaps divide the sentence into two.  

Line 461-462: I disagree with the statement that 16S- rRNA is among the most powerful tools to identify organisms to species level (to genus level yes). WGS would be a better method.

Line 492: Capital “T”0 in table 3.

Lines 507-509: What other evidence do you have to make this hypothesis?

Line 521: “Has” should be lower case.

Comments on the Quality of English Language

Overall the language was not an issue. A few minor corrections and rephrasing is enough.

Author Response

Dear reviewer,

I appreciate your time and valuable suggestion in reviewing our manuscript. Here is the correction to your comments and suggestion.

While I find the idea behind this study of relevance, I am not particularly satisfied with the story provided. The use of bioinformatics tools coupled with experimental validation are essential in creating links between what is predicted and what in reality happens. In this study the authors did a very good job experimentally to assess antimicrobial properties against certain pathogens but did not validate any of the NPs predicted by bioinformatics tools. Too many hypotheses with very little to back them up. I think that confirming at least 1 or 2 of the predictions would strengthen the manuscript substantially.

 Author response: We appreciate the suggestion and have made an additional attempt to identify the compounds to validate the experimental results using high resolution LC-QTOF-MS analysis. Briefly, the bacteria was re-grown in TSB media for 48 hours followed by ethyl acetate extraction. The sample was analyzed in LC-QTOF-MS and we were unable to identify any of the predicted compound from this study.

 We had made multiple attempts at bioassay-guided fractionation to identify the bioactive compound as well, but due to the instability or volatility of the compounds, the active compounds were lost consistently during the isolation process (Line 503-507). The reason could be the predicted BGCs or compounds in this study were based on the Bacterial WGS analysis and some microbes have cryptic/orphan BGCs that often need an external stimulation such as high temperature, or co-inoculation with other organisms etc to activate and produce the compounds. So, this could be the reason that these predicted compounds based on the WGS data were not present in our recent LC-QTOF-MS and lost in our bio-guided fractionation assay as well.

I would also restructure slightly the results section. Maybe create subsections for the use of the different tools so the reader doesn't get lost.

Author response: Completed. We gave subheadings to the result sections, where possible.

Minor corrections in the text and formatting issues:

Line 46: (e.g., Harpin Protein) (Chuang et al., 2014; Liu et al., 2020); and 46 Chitosan (Riseh et al., 2022). Shouldn’t the parentheses highlighted in red be placed at the of the sentence? 

Author response: Completed.

From what I understood, two examples are given form host resistance: Harpin protein and Chitosan.

Author response: Both Harpin and Chitosan are natural products derived from Bacteria and Crustacean, respectively. These compounds are not plant sourced but are well known to activate host resistance.

Page 9: The line numbers are overlapping with Table 1.

Author response: Completed

Figure 3 legend: “antimicro-bial” – remove the dash. Also, what are the numbers on the circus plot referring to? (e.g., 0.1). Please add the explanation to the legend.

Author response: Completed

Figure 4: Please add an explanation to the different numbers.

Author response: Explanation added to the legends of Fig 4.

 Supplementary table added as well: Table s1. A list of five global protein families (GPF) used for phylogenetic analysis is provided   as supplementary material.

Figure 4. A maximum likelihood phylogenetic tree illustrating the evolutionary relationship between Pseudomonas vancouverensis and other Pseudomonas species. Support values at each branch node represent confidence levels through 100 rounds of bootstrapping. The horizontal scale bar indicates the amount of genetic change, measured as nucleotide substitutions per site along the branches.

 Legend Figure 5: Font size too big. Please make fonts consistent.

Author response: Completed

Line 417-419: Phrase construction is a bit confusing (e.g. while … while). Perhaps divide the sentence into two.  

Author response: It was redundant, so deleted the redundant part.

Line 461-462: I disagree with the statement that 16S- rRNA is among the most powerful tools to identify organisms to species level (to genus level yes). WGS would be a better method.

Author response: I agree. However, species level confirmation is about 65-83% (Mignard and Flandrois 2006) in bacterial identification given the fact that the 16S rRNA gene (1,500 bp) is large enough for informatics purposes. Further, our phylogenetic tree analysis based on WGS data confirmed the bacterium identification as well.

Line 492: Capital “T”0 in table 3.

Author response: Fixed.

Lines 507-509: What other evidence do you have to make this hypothesis?

Author response: The sentence has been revised. We will work on to test the antifungal properties of some of the compounds predicted here.

Line 521: “Has” should be lower case.

Author response: Fixed.

Reviewer 2 Report

Comments and Suggestions for Authors

The manuscript addressed identification and analysis of biosynthetic gene clusters in Pseudomonas vancouverensis through whole-genome sequencing.  I appreciate the author's thorough approach and clear presentation of their nice piece of word. To further strengthen the manuscript, I recommend considering the following points:

1- The title can be amended to (Mining Pseudomonas vancouverensis for novel biosynthetic gene clusters using whole-genome sequencing).

2- Line 80: Specify the species names of Pseudomonas. Line 87: sp. not italic. Line 93, 186, 522: use abbreviation of Pseudomonas

3- Line 107:  Emphasize effectivelythe specific rationale/motivation for conducting this work.

4- Line 117: For how long did you incubate the bacterial isolate? 

5- Line 121: Add more details about the identification of the bacterial isolate such as the primers used for amplification the 16S rRNA gene and cite a relevant reference.

6- line 354: Pseudomonas should be italic.

7. Figure 4. all names should be italic. More details should be add such as the specific type of the phylogenetic tree, the accession numbers of the genes used, the meaning of the bar.

8- Adjust the font of Figure 5

9. line 547: Consider rephrasing such as  (Further study will determine the antifungal activity of putidacin against C. fragariae, B. cinerea, and P. obscurans).

10. Consider revise the citation style of all references as per the journal instruction. 

Comments on the Quality of English Language

Minor editing of English language required

Author Response

Dear reviewer,

We appreciate your time and valuable suggestion in reviewing our manuscript. Here is the correction to your comments and suggestion.

1- The title can be amended to (Mining Pseudomonas vancouverensis for novel biosynthetic gene clusters using whole-genome sequencing).

Author response: We changed the title as per your suggestion.

2- Line 80: Specify the species names of Pseudomonas. Line 87: sp. not italic. Line 93, 186, 522: use abbreviation of Pseudomonas

Author response:  We provided the Pseudomonas species name.

Spp. on line 87 is now regular (non-italic).

Line 93, 186, 522: Changed the Pseudomonas to abbreviation (P.)

3- Line 107:  Emphasize effectively the specific rationale/motivation for conducting this work.

Author response: We added the specific rationale for conducting this work.

4- Line 117: For how long did you incubate the bacterial isolate? 

Author response: Completed.

5- Line 121: Add more details about the identification of the bacterial isolate such as the primers used for amplification the 16S rRNA gene and cite a relevant reference.

Author response: Completed. I contacted the Azenta Life Sciences, but they said that is their proprietary material and they refused to provide the information.

6- line 354: Pseudomonas should be italic.

Author response: Completed

  1. Figure 4. all names should be italic. More details should be add such as the specific type of the phylogenetic tree, the accession numbers of the genes used, the meaning of the bar.

Author response:

Supplementary table added as well: Table s1. A list of five global protein families (GPF) used for phylogenetic analysis is provided   as supplementary material.

Figure 4. A maximum likelihood phylogenetic tree illustrating the evolutionary relationship between Pseudomonas vancouverensis and other Pseudomonas species. Support values at each branch node represent confidence levels through 100 rounds of bootstrapping. The horizontal scale bar indicates the amount of genetic change, measured as nucleotide substitutions per site along the branches.

8- Adjust the font of Figure 5

Author response: Font size has been changed.

  1. line 547: Consider rephrasing such as (Further study will determine the antifungal activity of putidacin against C. fragariae, B. cinerea, and P. obscurans).

Author response: We rephrased this sentence.

  1. Consider revise the citation style of all references as per the journal instruction. 

The citation has been revised to MDPI style.

Round 2

Reviewer 1 Report

Comments and Suggestions for Authors

Dear authors,

I am pleased with the modifications made and the responses given.

Good work.